# Composing Recurrent Spiking Neural Networks using Locally-Recurrent Motifs and Risk-Mitigating Architectural Optimization

## Abstract

In neural circuits, recurrent connectivity plays a crucial role in network function and stability. However, existing recurrent spiking neural networks (RSNNs) are often constructed by random connections without optimization. While RSNNs can produce rich dynamics that are critical for memory formation and learning, systemic architectural optimization of RSNNs is still an open challenge. We aim to enable systematic design of large RSNNs via a new scalable RSNN architecture and automated architectural optimization. We compose RSNNs based on a layer architecture called Sparsely-Connected Recurrent Motif Layer (SC-ML) that consists of multiple small recurrent motifs wired together by sparse lateral connections. The small size of the motifs and sparse inter-motif connectivity leads to an RSNN architecture scalable to large network sizes. We further propose a method called Hybrid Risk-Mitigating Architectural Search (HRMAS) to systematically optimize the topology of the proposed recurrent motifs and SC-ML layer architecture. HRMAS is an alternating two-step optimization process by which we mitigate the risk of network instability and performance degradation caused by architectural change by introducing a novel biologically-inspired "self-repairing" mechanism through intrinsic plasticity. The intrinsic plasticity is introduced to the second step of each HRMAS iteration and acts as unsupervised fast self-adaptation to structural and synaptic weight modifications introduced by the first step during the RSNN architectural "evolution". To the best of the authors' knowledge, this is the first work that performs systematic architectural optimization of RSNNs. Using one speech and three neuromorphic datasets, we demonstrate the significant performance improvement brought by the proposed automated architecture optimization over existing manually-designed RSNNs.

## 1 Introduction

In the brain, recurrent connectivity is indispensable for maintaining dynamics, functions, and oscillations of the network (Buzsaki, 2006). As a brain-inspired computational model, spiking neural networks (SNNs) are well suited for processing spatiotemporal information (Maass, 1997). In particular, recurrent spiking neural networks (RSNNs) can mimic microcircuits in the biological brain and induce rich behaviors that are critical for memory formation and learning. Recurrence has been explored in conventional non-spiking artificial neural networks (ANNs) in terms of Long Short Term Memory (LSTM) (Hochreiter & Schmidhuber, 1997), Echo State Networks (ESN) (Jaeger, 2001), Deep RNNs (Graves et al., 2013), Gated Recurrent Units (GRU) (Cho et al., 2014), and Legendre Memory Units (LMU) (Voelker et al., 2019). While recurrence presents unique challenges and opportunities in the context of spiking neural networks, RSNNs are yet to be well explored.

Most existing works on RSNNs adopt recurrent layers or reservoirs with randomly generated connections. The Liquid State Machine (LSM) (Maass et al., 2002) is one of the most widely adopted RSNN architectures with one or multiple recurrent reservoirs and an output readout layer wired up using feedforward synapses (Zhang et al., 2015; Wang & Li, 2016; Srinivasan et al., 2018). However, there is a lack of principled approaches for setting up the recurrent connections in reservoirs. Instead, ad-hoc randomly generated wiring patterns are often adopted. Bellec et al. (2018) proposed an architecture called long short-term memory SNNs (LSNNs). The recurrent layer contains a regular

spiking portion with both inhibitory and excitatory spiking neurons and an adaptive neural population. Zhang & Li (2019b) proposed to train deep RSNNs by a spike-train level backpropagation (BP) method. Maes et al. (2020) demonstrated a new reservoir with multiple groups of excitatory neurons and a central group of inhibitory neurons. Furthermore, Zhang & Li (2020a) presented a recurrent structure named ScSr-SNNs in which recurrence is simply formed by a self-recurrent connection to each neuron. However, the recurrent connections in all of these works are either randomly generated with certain probabilities or simply constructed by self-recurrent connections. Randomly generated or simple recurrent connections may not effectively optimize RSNNs' performance. Recently, Pan et al. (2023) introduced a multi-objective Evolutionary Liquid State Machine (ELSM) inspired by neuroevolution process. Chakraborty & Mukhopadhyay (2023) proposed Heterogeneous recurrent spiking neural network (HRSNN), in which recurrent layers are composed of heterogeneous neurons with different dynamics. Chen et al. (2023) introduced an intralayer-connected SNN and a hybrid training method combining probabilistic spike-timing dependent plasticity (STDP) with BP. But their performance still has significant gaps. Systemic RSNN architecture design and optimization remain as an open problem.

Neural architectural search (NAS), the process of automating the construction of non-spiking ANNs, has become prevalent recently after achieving state-of-the-art performance on various tasks (Elsken et al., 2019; Wistuba et al., 2019). Different types of strategies such as reinforcement learning (Zoph & Le, 2017), gradient-based optimization (Liu et al., 2018), and evolutionary algorithms (Real et al., 2019) have been proposed to find optimal architectures of traditional CNNs and RNNs. In contrast, the architectural optimization of SNNs has received little attention. Only recently, Tian et al. (2021) adopted a simulated annealing algorithm to learn the optimal architecture hyperparameters of liquid state machine (LSM) models through a three-step search. Similarly, a surrogate-assisted evolutionary search method was applied in Zhou et al. (2020) to optimize the hyperparameters of LSM such as density, probability and distribution of connections. However, both studies focused only on LSM for which hyperparameters indirectly affecting recurrent connections as opposed to specific connectivity patterns were optimized. Even after selecting the hyperparameters, the recurrence in the network remained randomly determined without any optimization. Recently, Kim et al. (2022) explored a cell-based neural architecture search method on SNNs, but did not involve large-scale recurrent connections. Na et al. (2022) introduced a spike-aware NAS framework called AutoSNN to investigate the impact of architectural components on SNNs' performance and energy efficiency. Overall, NAS for RSNNs is still rarely explored.

This paper aims to enable systematic design of large recurrent spiking neural networks (RSNNs) via a new scalable RSNN architecture and automated architectural optimization. RSNNs can create complex network dynamics both in time and space, which manifests itself as an opportunity for achieving great learning capabilities and a challenge in practical realization. It is important to strike a balance between theoretical computational power and architectural complexity. Firstly, we argue that composing RSNNs based on well-optimized building blocks small in size, or recurrent motifs, can lead to an architectural solution scalable to large networks while achieving high performance. We assemble multiple recurrent motifs into a layer architecture called Sparsely-Connected Recurrent Motif Layer (SC-ML). The motifs in each SC-ML share the same *topology*, defined by the size of the motif, i.e., the number of neurons, and the recurrent connectivity pattern between the neurons. The motif topology is determined by the proposed architectural optimization while the weights within each motif may be tuned by standard backpropagation training algorithms. Motifs in a recurrent SC-ML layer are wired together using sparse lateral connections determined by imposing spatial connectivity constraints. As such, there exist two levels of structured recurrence: recurrence within each motif and recurrence between the motifs at the SC-ML level. The fact that the motifs are small in size and that inter-motif connectivity is sparse alleviates the difficulty in architectural optimization and training of these motifs and SC-ML. Furthermore, multiple SC-ML layers can be stacked and wired using additional feedforward weights to construct even larger recurrent networks.

Secondly, we demonstrate a method called Hybrid Risk-Mitigating Architectural Search (HRMAS) to optimize the proposed recurrent motifs and SC-ML layer architecture. HRMAS is an alternating two-step optimization process hybridizing bio-inspired intrinsic plasticity for mitigating the risk in architectural optimization. Facilitated by gradient-based methods (Liu et al., 2018; Zhang & Li, 2020b), the first step of optimization is formulated to optimize network architecture defined by the size of the motif, intra and inter-motif connectivity patterns, types of these connections, and the corresponding synaptic weight values, respectively.

While structural changes induced by the architectural-level optimization are essential for finding high-performance RSNNs, they may be misguided due to discontinuity in architectural search, and limited training data, hence leading to over-fitting. We mitigate the risk of network instability and performance degradation caused by architectural change by introducing a novel biologically-inspired "self-repairing" mechanism through intrinsic plasticity, which has the same spirit of homeostasis during neural development (Tien & Kerschensteiner, 2018). The intrinsic plasticity is introduced in the second step of each HRMAS iteration and acts as unsupervised self-adaptation to mitigate the risks imposed by structural and synaptic weight modifications introduced by the first step during the RSNN architectural "evolution".

We evaluate the proposed techniques on speech dataset TI46-Alpha (Liberman et al., 1991), neuromorphic speech dataset N-TIDIGITS (Anumula et al., 2018), neuromorphic video dataset DVS-Gesture (Amir et al., 2017), and neuromorphic image dataset N-MNIST (Orchard et al., 2015). The SC-ML-based RSNNs optimized by HRMAS achieve state-of-the-art performance on all four datasets. With the same network size, automated network design via HRMAS outperforms existing RSNNs by up to 3.38% performance improvement.

## 2    SPARSELY-CONNECTED RECURRENT MOTIF LAYER (SC-ML)

Unlike the traditional non-spiking RNNs that are typically constructed with units like LSTM or GRU, the structure of existing RSNNs is random without specific optimization, which hinders RSNNs' performance and prevents scaling to large networks. However, due to the complexity of recurrent connections and dynamics of spiking neurons, the optimization of RSNNs weights is still an open problem. As shown in Table 2, recurrent connections that are not carefully set up may hinder network performance. To solve this problem, we first designed the SC-ML layer, which is composed of multiple sparsely-connected recurrent *motifs*, where each motif consists of a group of recurrently connected spiking neurons, as shown in Figure 1. The motifs in each SC-ML share the same topology, which is defined as the size of the motif, i.e., the number of neurons, and the recurrent connectivity pattern between the neurons (excitatory, inhibitory or non-existent). Within the motif, synaptic connections can be constructed between any two neurons including self-recurrent connections. Thus the problem of the recurrent layer optimization can be simplified to that of learning the optimal motif and sparse inter-motif connectivity, alleviating the difficulty in architectural optimization and allowing scalability to large networks.

This motif-based structure is motivated by both a biological and a computational perspective. First, from a biological point of view, there is evidence that the neocortex is not only organized in layered minicolumn structures but also into synaptically connected clusters of neurons within such structures (Perin et al., 2011; Ko et al., 2011). For example, the networks of pyramidal cells cluster into multiple groups of a few dozen neurons each. Second, from a computational perspective, optimizing the connectivity of the basic building block, i.e., the motif, simplifies the problem of optimizing the connectivity of the whole recurrent layer. Third, by constraining most recurrent connections inside the

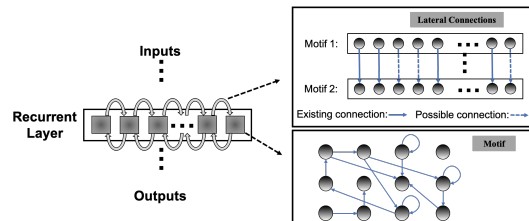

Figure 1: Sparsely-Connected Recurrent Motif Layer.

motifs and allowing a few lateral connections between neighboring motifs to exchange information across the SC-ML, the total number of recurrent connections is limited. This leads to a great deal of sparsity as observed in biological networks (Seeman et al., 2018).

Figure 1 presents an example of SC-ML with 12-neuron motifs. The lateral inter-motif connections can be introduced as the mutual connections between two corresponding neurons in neighboring motifs to ensure sparsity and reduce complexity. With the proposed SC-ML, one can easily stack multiple SC-MLs to form a multi-layer large RSNN using feedforward weights. Within a multi-layered network, information processing is facilitated through local processing of different motifs, communication of motif-level responses via inter-motif connections, and extraction and processing of higher-level features layer by layer.

## 3  HYBRID RISK-MITIGATING ARCHITECTURAL SEARCH (HRMAS)

To enhance the performance of RSNNs, we introduce the Hybrid Risk-Mitigating Architectural Search (HRMAS). This framework systematically optimizes the motif topology and lateral connections of SC-ML. Each optimization iteration consists of two alternating steps.

### 3.1  HYBRID RISK-MITIGATING ARCHITECTURAL SEARCH FRAMEWORK

In HRMAS, all recurrent connections are categorized into three types: inhibitory, excitatory, and non-existence. An inhibitory connection has a negative weight and is fixed without training in our current implementation, similar to the approach described in (Zhang & Li, 2020a; 2021). The weight of an excitatory connection is positive and trained by a backpropagation (BP) method. HRMAS is an alternating two-step optimization process, hybridizing architectural optimization with intrinsic plasticity (IP). The first step of each HRMAS optimization iteration optimizes the topology of the motif and inter-motif connectivity in SC-ML and the corresponding synaptic weights hierarchically. Specifically, the optimal number of neurons in the motif is optimized over a finite set of motif sizes. All possible intra-motif connections are considered and the type of each connection is optimized, which may lead to a sparser connectivity if the connection types of certain synapses are determined to be "non-existence". At the inter-motif level, a sparse motif-to-motif connectivity constraint is imposed: neurons in one motif are only allowed to be wired up with the corresponding neurons in the neighboring motifs. Inter-motif connections also fall under one of the three types. Hence, a greater level of sparsity is produced with the emergence of connections of type "non-existence". The second step in each HRMAS iteration executes an unsupervised IP rule to stabilize the network function and mitigate potential risks caused by architectural changes.

Figure 2 illustrates the incremental optimization strategy we adopt for the architectural parameters. Using the two-step optimization, initially all architectural parameters including motif size and connectivity are optimized. After several training iterations, we choose the optimal motif size from a set of discrete options. As the most critical architectural parameter is set, we continue to optimize the remaining architectural parameters defining connectivity, allowing fine-tuning of performance based on the chosen motif size.

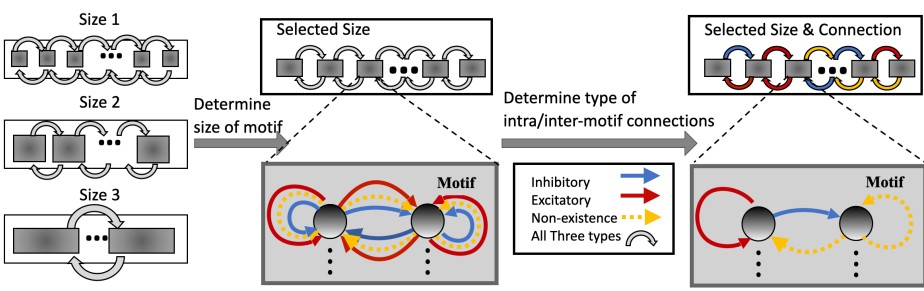

Figure 2: Architectural optimization in HRMAS.

### 3.1.1  COMPARISON WITH PRIOR NEURAL ARCHITECTURAL SEARCH WORK OF NON-SPIKING RNNS

Neural architecture search (NAS) has been applied for architectural optimization of traditional non-spiking RNNs, where a substructure called cell is optimized by a search algorithm (Zoph & Le, 2017). Nevertheless, this NAS approach may not be the best fit for RSNNs. First, recurrence in the cell is only created by feeding previous hidden state back to the cell while connectivity inside the cell is feedforward. Second, the overall operations and connectivity found by the above NAS procedure do not go beyond an LSTM-like architecture. Finally, the considered combination operations and activation functions like addition and elementwise multiplication are not biologically plausible.

In comparison, in RSNNs based on the proposed SC-ML architecture, we add onto the memory effects resulting from temporal integration of individual spiking neurons by introducing sparse intra

or inter-motif connections. This corresponds to a scalable and biologically plausible RSNN architectural design space that closely mimics the microcircuits in the nervous system. Furthermore, we develop the novel alternating two-step HRMAS framework hybridizing gradient-based optimization and biologically-inspired intrinsic plasticity for robust NAS of RSNNs.

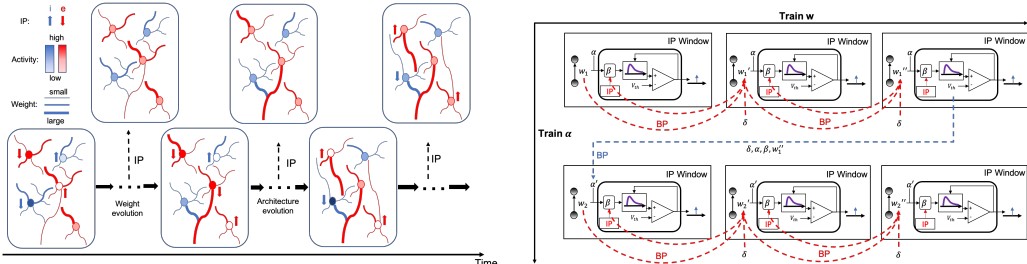

Figure 3: Evolution in neural development.

Figure 4: Proposed HRMAS.

## 3.2 Alternating Two-Step Optimization in HRMAS

The alternating two-step optimization in HRMAS is inspired by the evolution in neural development. As shown in Figure 3, neural circuits may experience weight changes through synaptic plasticity. Over a longer time scale, circuit architecture, i.e., connectivity, may evolve through learning and environmental changes. In addition, spontaneous firing behaviors of individual neurons may be adapted by intrinsic plasticity (IP). We are motivated by the important role of local IP mechanisms in stabilizing neuronal activity and coordinating structural changes to maintain proper circuit functions (Tien & Kerschensteiner, 2018). We view IP as a "fast-paced" self-adapting mechanism of individual neurons to react to and minimize the risks of weight and architectural modifications. As shown in Figure 4, we define the architectural parameters (motif size and intra/inter-motif connection types weights), synaptic weights, and intrinsic neuronal parameters as $\alpha$, $w$, and $\beta$, respectively. Each HRMAS optimization iteration consists of two alternating steps. In the first step, we optimize $\alpha$ and $w$ hierarchically based on gradient-based optimization using backpropagation (BP). In Figure 4, $\delta$ is the backpropagated error obtained via the employed BP method. In the second step, we use an unsupervised IP rule to adapt the intrinsic neuronal parameters of each neuron over a time window ("IP window") during which training examples are presented to the network. IP allows the neurons to respond to the weight and architectural changes introduced in the first step and mitigate possible risks caused by such changes. In Step 1 of the subsequent iteration, the error gradients w.r.t the synaptic weights and architectural parameters are computed based on the most recent values of $\beta$ updated in the preceding iteration. In summary, the $k$-th HRMAS iteration solves a bi-level optimization problem:

$$\alpha^* = \arg\min_{\alpha} \mathcal{L}_{\text{valid}}(\alpha, w^*(\alpha), \beta^*) \tag{1}$$

$$\text{s.t.} \quad \beta^* = \arg\min_{\beta} \mathcal{L}_{\text{ip}}(\alpha, w^*(\alpha), \beta^*_-), \tag{2}$$

$$\text{s.t.} \quad w^*(\alpha) = \arg\min_{w} \mathcal{L}_{\text{train}}(\alpha, w, \beta^*_-), \tag{3}$$

where $\mathcal{L}_{valid}$ and $\mathcal{L}_{train}$ are the loss functions defined based on the validation and training sets used to train $\alpha$ and $w$ respectively; $\mathcal{L}_{ip}$ is the local loss to be minimized by the IP rule as further discussed in Section 3.2.2; $\beta^*_-$ are the intrinsic parameter values updated in the preceding $(k-1)$-th iteration; $w^*(\alpha)$ denotes the optimal synaptic weights under the architecture specified by $\alpha$. The complete derivation of the proposed optimization techniques can be found in the Supplemental Material.

### 3.2.1 Gradient-based Optimization in HRMAS

Optimizing the weight and architectural parameters by solving the bi-level optimization problem of (1, 2, 3) can be computationally expensive. We adapt the recent method proposed in Liu et al. (2018) to reduce computational complexity by relaxing the discrete architectural parameters to continuous ones for efficient gradient-based optimization. Without loss of generality, we consider a multi-layered RSNN consisting of one or more SC-ML layers, where connections between layers are

assumed to be feedforward. We focus on one SC-ML layer, as shown in Figure 5, to discuss the proposed gradient-based optimization.

The number of neurons in the SC-ML layer is fixed. The motif size is optimized such that each neuron is partitioned into a specific motif based on the chosen motif size. The largest white square in Figure 5 shows the layer-connectivity matrix of all intra-layer connections of the whole layer, where the dimension of the matrix corresponds to the neuron count of the layer. We superimpose three sets of smaller gray squares onto the layer-connectivity matrix, one for each of the three possible motif sizes of $v_1$, $v_2$, and $v_3$ considered. Choosing a particular motif size packs neurons in the layer into multiple motifs, and the corresponding gray squares illustrate the intra-motif connectivity introduced within the SC-ML layer.

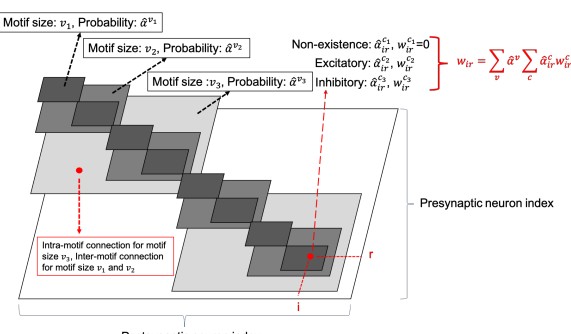

Figure 5: SC-ML with relaxed architectural parameters.

The entry of the layer-connectivity matrix at row $r$ and column $i$ specifies the existence and nature of the connection from neuron $r$ to neuron $i$. We consider multiple motif size and connection type choices during architectural search using continuous-valued parameterizations $\alpha^v$ and $\alpha_{ir}^c$, respectively for each motif size $v$ and connection type $c$. We relax the categorical choice of each motif size using a softmax over all possible options: $\hat{\alpha}^v = \frac{exp(\alpha^v)}{\sum_{v' \in \mathcal{V}} exp(\alpha^{v'})}$, and similarly relax the categorical choice of each connection type based on the corresponding motif size: $\hat{\alpha}_{ir}^c = \frac{exp(\alpha_{ir}^c)}{\sum_{c' \in \mathcal{C}} exp(\alpha_{ir}^{c'})}$. Here, $\mathcal{C}$ and $\mathcal{V}$ are the set of all possible connection types and motif sizes, respectively; $\hat{\alpha}^v$ and $\hat{\alpha}_{ir}^c$ are the continuous-valued categorical choice of motif size $v$ and connection type $c$, respectively, which can also be interpreted as the probability of selecting the corresponding motif size or connection type. As in Figure 5, the synaptic weight of the connection from neuron $r$ to neuron $i$ is expressed as the summation of weights under all possible motif sizes and connection types weighted by the respective continuous-valued categorical choices (selection probabilities).

Based on the leaky integrate-and-fire (LIF) neuron model (Gerstner & Kistler, 2002), the neuronal membrane voltage $u_i[t]$ of neuron $i$ in the SC-ML layer at time $t$ is given by integrating currents from all inter-layer inputs and intra-layer recurrent connections under all possible architectural parameterizations:

$$u_i[t] = (1 - \frac{1}{\tau})u_i[t-1] + \frac{R}{\tau}(\sum_j w_{ij}a_j[t] + \sum_{v \in \mathcal{V}}(\hat{\alpha}^v \sum_r^{I_i^v} \sum_{c \in \mathcal{C}} (\hat{\alpha}_{ir}^c w_{ir}^c a_r[t-1]))), \qquad (4)$$

where $R$ and $\tau$ are the resistance and time constant of the membrane, $w_{ij}$ the synaptic weight from neuron $j$ in the previous layer to neuron $i$, $w_{ir}^c$ the recurrent weight from neuron $r$ to neuron $i$ of connection type $c$, and $a_j[t]$ the (unweighted) postsynaptic current (PSC) converted from spikes of neuron $j$ through a synaptic model. To reduce clutter in the notation, we use $I_i^v$ to denote the number of presynaptic connections afferent onto neuron $i$'s input in the recurrent layer when choosing motif size $v$, which includes both inter-motif and intra-motif connections. We further drop the explicit dependence of $\hat{\alpha}_{ir}^c$ on $\hat{\alpha}^v$. Through (4), the continuous architecture parameterizations influence the integration of input currents, and hence firing activities of neurons in all layers and affect the loss function defined at the output layer. As such, the task of architecture optimization reduces to the one that learns the set of optimal continuous variables $\hat{\alpha}^c$ and $\hat{\alpha}^v$. The final architecture is constructed by choosing the parameterizations with the highest selection probabilities obtained from the optimization.

We solve the bi-level optimization defined in (1), (2), (3) using the Temporal Spike Sequence Learning via Backpropagation (TSSL-BP) method (Zhang & Li, 2020b), which handles non-differentiability of the all-or-none spiking neural activation function. To alleviate the computational overhead, we approximate $w^*(\alpha)$ in (3) by a one step of gradient-based update: $w^*(\alpha) \approx$

$w - \eta \nabla_w \mathcal{L}_{train}(w, \alpha, \beta_-^*)$, where $w$ are the initial weight values. The weights and architectural parameters are updated by gradient descent as:

$$\Delta w_{ij} \propto \delta_i[t] \frac{R}{\tau} a_j[t], \quad \Delta \hat{\alpha}^v \propto \sum_i^{N_r} \delta_i[t] \frac{R}{\tau} \sum_r^{I_i^v} (\sum_{c \in \mathcal{C}} \hat{\alpha}_{ir}^c w_{ir}^c a_r[t-1]),$$

$$\Delta w_{ir}^c \propto \delta_i[t] \frac{R}{\tau} \sum_{v \in \mathcal{V}} (\hat{\alpha}^v \hat{\alpha}_{ir}^c a_r[t-1]), \quad \Delta \hat{\alpha}_{ir}^c \propto \delta_i[t] \frac{R}{\tau} \sum_{v \in \mathcal{V}} (\hat{\alpha}^v w_{ir}^c a_r[t-1]). \tag{5}$$

where $\delta_i[t]$ is the backpropagated error for neuron $i$ at time $t$ given in (22) of the Supplemental Material, $N_r$ is the number of neurons in this recurrent layer, $R$ and $\tau$ are the leaky resistance and membrane time constant, two intrinsic parameters adapted by the IP rule, $a_j[t]$ and $a_r[t]$ are the (unweighted) postsynaptic currents (PSCs) generated based on synaptic model by the presynaptic neuron $j$ in the preceding layer and the $r$-th neuron in this recurrent layer, respectively. We include all details of the proposed gradient-based method and derivation of the involved error backpropagation in Section B and Section C of the Supplementary Material.

### 3.2.2 RISK MINIMIZING OPTIMIZATION WITH INTRINSIC PLASTICITY

For architectural optimization of non-spiking RNNs, gradient-based methods are shown to be unstable in some cases due to misguided architectural changes and conversion from the optimized continuous-valued parameterization to a discrete architectural solution, hindering the final performance and demolishing the effectiveness of learning (Zela et al., 2019). Adaptive regularization which modifies the regularization strength (weight decay) guided by the largest eigenvalue of $\nabla_\alpha^2 \mathcal{L}_{valid}$ was proposed to address this problem (Zela et al., 2019). While this method shows promise for non-spiking RNNs, it is computationally intensive due to frequent expensive eigenvalue computation, severely limiting its scalability.

To address risks observed in architectural changes for RSNNs, we introduce a biologically-inspired risk-mitigation method. Biological circuits demonstrate that Intrinsic Plasticity (IP) is crucial in reducing such risks. IP is a self-regulating mechanism in biological neurons ensuring homeostasis and influencing neural circuit dynamics (Marder et al., 1996; Baddeley et al., 1997; Desai et al., 1999). It not only stabilizes neuronal activity but also coordinates connectivity and excitability changes across neurons to stabilize circuits (Maffei & Fontanini, 2009; Tien & Kerschensteiner, 2018). Drawing from these findings, our HRMAS framework integrates the IP rule into the architectural optimization, applied in the second step of each iteration. IP is based on local neural firing activities and performs online adaptation with minimal additional computational overhead.

IP has been applied in spiking neural networks for locally regulating neuron activity (Lazar et al., 2007; Bellec et al., 2018). In this work, we make use of IP for mitigating the risk of RSNN architectural modifications. We adopt the SpiKL-IP rule (Zhang & Li, 2019a) for all recurrent neurons during architecture optimization. SpiKL-IP adapts the intrinsic parameters of a spiking neuron while minimizing the KL-divergence from the output firing rate distribution to a targeted exponential distribution. It both maintains a level of network activity and maximizes the information transfer for each neuron. We adapt leaky resistance and membrane time constant of each neuron using SpiKL-IP which effectively solves the optimization problem in (2) in an online manner. The proposed alternating two-step optimization of HRMAS is summarized in Algorithm 1. More details of the IP implementation can be found in Section D of the Supplementary Material.

---

**Algorithm 1:** HRMAS - Hybrid Risk-Mitigating Architectural Search

---

Initialize weights $w$, intrinsic parameters $\beta$, architectural parameters $\alpha$, and correspondingly $\hat{\alpha}$.

**while** *no converged* **do**

    Update $\hat{\alpha}$ by $\eta_1 \nabla_{\hat{\alpha}} \mathcal{L}_{valid}(\hat{\alpha}, w - \eta_2 \nabla_w \mathcal{L}_{train}(\hat{\alpha}, w, \beta))$;

    Update $w$ by $\eta_2 \nabla_w \mathcal{L}_{train}(\hat{\alpha}, w, \beta)$;

    $\beta \longleftarrow$ SpiKL-IP$(\hat{\alpha}, w)$

**end**

---

## 4 EXPERIMENTAL RESULTS

The proposed HRMAS optimized RSNNs with the SC-ML layer architecture and five motif size options are evaluated on speech dataset TI46-Alpha (Liberman et al., 1991), neuromorphic speech dataset N-TIDIGITS (Anumula et al., 2018), neuromorphic video dataset DVS-Gesture (Amir et al., 2017), and neuromorphic image dataset N-MNIST (Orchard et al., 2015). The performances are compared with recently reported state-of-the-art manually designed architectures of SNNs and ANNs such as feedforward SNNs, RSNNs, LSM, and LSTM. The details of experimental settings, hyperparameters, loss function and dataset preprocessing are described in Section E of the Supplementary Material. For the proposed work, the architectural parameters are optimized by HRMAS with the weights trained on a training set and architectural parameters learned on a validation set as shown in Algorithm 1. The accuracy of each HRMAS optimized network is evaluated on a separate testing set with all weights reinitialized. Table 1 shows all results.

Table 1: Accuracy on TI46-Alpha, N-TIDIGITS, DVS-Gesture and N-MNIST.

| Network Structure | Learning Rule | Hidden Layers | Best |
|---|---|---|---|
| TI46-Alpha | | | |
| LSM (Wijesinghe et al., 2019) | Non-spiking BP | 2000 | 78% |
| RSNN (Zhang & Li, 2019b) | ST-RSBP | $400 - 400 - 400$ | 93.35% |
| Sr-SNN (Zhang & Li, 2020a) | TSSL-BP | $400 - 400 - 400$ | 94.62% |
| This work | TSSL-BP | 800 | **96.44%** |
| N-TIDIGITS | | | |
| GRU (Anumula et al., 2018) | Non-spiking BP | $200 - 200 - 100$ | 90.90% |
| Phase LSTM (Anumula et al., 2018) | Non-spiking BP | $250 - 250$ | 91.25% |
| RSNN (Zhang & Li, 2019b) | ST-RSBP | $400 - 400 - 400$ | 93.90% |
| Feedforward SNN | TSSL-BP | 400 | 84.84% |
| This work | TSSL-BP | 400 | **94.66%** |
| DVS-Gesture | | | |
| Feedforward SNN (He et al., 2020) | STBP | $P4 - 512$ | 87.50% |
| LSTM (He et al., 2020) | Non-spiking BP | $P4 - 512$ | 88.19% |
| HeNHeS (Chakraborty & Mukhopadhyay, 2023) | STDP | 500 | 90.15% |
| Feedforward SNN | TSSL-BP | $P4 - 512$ | 88.19% |
| This work | TSSL-BP | $P4 - 512$ | **90.28%** |
| N-MNIST | | | |
| Feedforward SNN (He et al., 2020) | STBP | 512 | 98.19% |
| RNN (He et al., 2020) | Non-spiking BP | 512 | 98.15% |
| LSTM (He et al., 2020) | Non-spiking BP | 512 | 98.69% |
| ELSM(Pan et al., 2023) | Non-spiking BP | 8000 | 97.23% |
| This work | TSSL-BP | 512 | **98.72%** |

### 4.1 RESULTS

Table 1 shows the results on the TI46-Alpha dataset. The HRMAS-optimized RSNN has one hidden SC-ML layer with 800 neurons, and outperforms all other models while achieving 96.44% accuracy with mean of 96.08% and standard deviation (std) of 0.27% on the testing set. The proposed RSNN outperforms the LSM model in Wijesinghe et al. (2019) by 18.44%. It also outperforms the larger multi-layered RSNN with more tunable parameters in Zhang & Li (2019b) trained by the spike-train level BP (ST-RSBP) by 3.1%. Recently, Zhang & Li (2020a) demonstrated improved performances from manually designed RNNs with self-recurrent connections trained using the same TSSL-BP method. Our automated HRMAS architectural search also produces better performing networks.

We also show that a HRMAS-optimized RSNN with a 400-neuron SC-ML layer outperforms several state-of-the-art results on the N-TIDIGITS dataset (Zhang & Li, 2019b), achieving 94.66% testing accuracy (mean: 94.27%, std: 0.35%). Our RSNN has more than a 3% performance gain over the widely adopted recurrent structures of ANNs, the GRU and LSTM. It also significantly outperforms

a feedforward SNN with the same hyperparameters, achieving an accuracy improvement of almost 9.82%, demonstrating the potential of automated architectural optimization.

On DVS-Gesture and N-MNIST, our method achieves accuracies of 90.28% (mean: 88.40%, std: 1.71%) and 98.72% (mean: 98.60%, std: 0.08%), respectively. Table1 compares a HRMAS-optimized RSNN with models including feedforward SNNs trained by TSSL-BP (Zhang & Li, 2020b) or STBP (Wu et al., 2018) with the same size, and non-spiking ANNs vanilla LSTM (He et al., 2020). Note that although our RSNN and the LSTM model have the same number of units in the recurrent layer, the LSTM model has a much greater number of tunable parameters and a improved rate-coding-inspired loss function. Our HRMAS-optimized model surpasses all other models. For a more intuitive understanding, Figure 6 presents two examples of the motif topology optimized by HRMAS: motif sizes 2 in options $[2, 4, 8, 16, 32]$ for the N-MNIST dataset and motif size 16 in options $[5, 10, 16, 25, 40]$ for the TI-Alpha dataset.

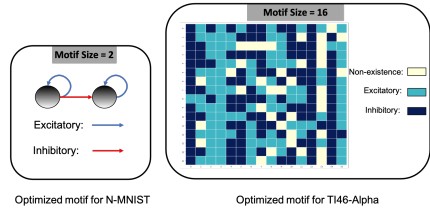

Figure 6: Optimized motif topologies.

## 4.2 ABLATION ANALYSIS

We conduct ablation studies on the RSNN optimized by HRMAS for the TI46-Alpha dataset to reveal the contributions of various proposed techniques. When all proposed techniques are included, the HRMAS-optimized RSNN achieves 96.44% accuracy. In Table 2, removing of the IP rule from the second step of the HRMAS optimization iteration visibly degrades the performance, showing the efficacy of intrinsic plasticity for mitigating risks of architectural changes. A similar performance degradation is observed when the sparse inter-motif connections are excluded from the SC-ML layer architecture. Without imposing a structure in the hidden layer by using motifs as a basic building block, HRMAS can optimize all possible connectivity types of the large set of 800 hidden neurons. However, this creates a large and highly complex architectural search space, rendering a tremendous performance drop. Finally, we compare the HRMAS model with an RSNN of a fixed architecture with full recurrent connectivity in the hidden layer. The application of the BP method is able to train the latter model since no architectural (motifs or connection types) optimization is involved. However, albeit its significantly increased model complexity due to dense connections, this model has a large performance drop in comparison with the RSNN fully optimized by HRMAS.

Table 2: Ablation studies of HRMAS on TI46-Alpha

| Setting | Accuracy | Setting | Accuracy |
|---|---|---|---|
| Without IP | 95.20% | Without inter-motif connections | 95.73% |
| Without motif | 88.35% | Fully connected RSNN | 94.10% |

## 5 CONCLUSION

We present an RSNN architecture based on SC-ML layers composed of multiple recurrent motifs with sparse inter-motif connections as a solution to constructing large recurrent spiking neural models. We further propose the automated architectural optimization framework HRMAS hybridizing the "evolution" of the architectural parameters and corresponding synaptic weights based on back-propagation and biologically-inspired mitigation of risks of architectural changes using intrinsic plasticity. We show that HRMAS-optimized RSNNs impressively improve performance on four datasets over the previously reported state-of-the-art RSNNs and SNNs. Notably, our HRMAS framework can be easily extended to more flexible network architectures, optimizing sparse and scalable RSNN architectures. By sharing the PyTorch implementation of our HRMAS framework, this work aims to foster advancements in high-performance RSNNs for both general-purpose and dedicated neuromorphic computing platforms, potentially inspiring innovative designs in brain-inspired recurrent spiking neural models and their energy-efficient deployment.

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

APPENDIX: SUPPLEMENTARY MATERIALS

## A    SPIKING NEURON MODEL

In this work, we adopt the leaky integrate-and-fire (LIF) neuron model (Gerstner & Kistler, 2002) which is one of the most popular neuron models for simulating SNNs. During the simulation, we use the fixed-step first-order Euler method to discretize the LIF model. In the rest of this paper, we only analyze an SNN in the discretized form.

Consider the input spike train from pre-synaptic neuron $j$: $s_j[t] = \sum_{t_j^{(f)}} \delta[t - t_j^{(f)}]$, where $t_j^{(f)}$ denotes a particular firing time of presynaptic neuron $j$. The incoming spikes are converted into an (unweighted) postsynaptic current (PSC) $a_j[t]$ through a synaptic model. We adopt the first-order synaptic model (Gerstner & Kistler, 2002):

$$a_j[t] = (1 - \frac{1}{\tau_{syn}})a_j[t-1] + s_j[t], \tag{6}$$

where $\tau_{syn}$ is the synaptic time constant.

Then, the neuronal membrane voltage $u_i[t]$ of neuron $i$ at time $t$ is given by

$$u_i[t] = (1 - \frac{1}{\tau})u_i[t-1] + \frac{R}{\tau}\sum_j w_{ij}a_j[t], \tag{7}$$

where $R$ and $\tau$ are the resistance and time constant of the membrane, $w_{ij}$ the synaptic weight from pre-synaptic neuron $j$ to neuron $i$. Moreover, the firing output of the neuron is expressed as

$$s_i[t] = H\left(u_i[t] - V_{th}\right), \tag{8}$$

where $V_{th}$ is the firing threshold and $H(\cdot)$ is the Heaviside step function.

## B    GRADIENT-BASED OPTIMIZATION ON ARCHITECTURAL PARAMETERS

In the proposed HRMAS, architectural parameters and synaptic weights are optimized by the first step. The architectural parameters are defined as motif size and types of intra/inter-motif connections. The general architectural optimization is performed by generating architecture and evaluating the architecture by a standard training and validation process on data. The validation performance is used to train the architectural parameters and generate a better structure. These steps are repeated until the optimal architecture is found. The first step of the $k$-th HRMAS iteration solves a bi-level optimization problem using BP:

$$min_\alpha \mathcal{L}_{valid}(\alpha, w^*(\alpha), \beta_-^*) \tag{9}$$
$$s.t.\quad w^*(\alpha) = arg_w min\mathcal{L}_{train}(\alpha, w, \beta_-^*), \tag{10}$$

where $\mathcal{L}_{valid}$ and $\mathcal{L}_{train}$ are the loss functions defined based on the validation and training sets used to train $\alpha$ and $w$ respectively; $\beta_-^*$ is the intrinsic parameter values updated in the preceding $(k-1)$-th iteration; $w^*(\alpha, \beta)$ denotes the optimal synaptic weights under the architecture specified by $\alpha$. The second step of the $k$-th iteration solves the optimization problem below:

$$\beta^* = arg_\beta min\mathcal{L}_{ip}(\alpha^*, w^*, \beta) \tag{11}$$

$\mathcal{L}_{ip}$ is the local loss to be minimized by the IP rule. Since the synaptic weights and architectural parameters are computed based on the most recent values of neuronal parameters updated in its preceding iteration, the IP rule applied in the second step of HRMAS iteration is independent of this bi-level optimization problem. Therefore, in this section, we do not express the IP method explicitly and define the bi-level optimization problem as

$$min_\alpha \mathcal{L}_{valid}(\alpha, w^*(\alpha))$$
$$s.t.\quad w^*(\alpha) = arg_w min\mathcal{L}_{train}(\alpha, w), \tag{12}$$

Solving bi-level optimization problem has intensive demand on computational resources because it usually requires the validation performance of all the intermediate architecture. Recently, a gradient-based search method (Liu et al., 2018) is proposed focusing on efficient architecture search. It

significantly reduces the computational cost of architecture search by approximating the bi-level optimization problem, relaxing the discrete architectural parameters to continuous ones, and solving the continuous model by gradient descent. In the proposed HRMAS framework, we also adopt the gradient-based approach together with the backpropagation method of SNNs to optimize the SC-ML architecture.

We denote the type of connection as $c$, and the size of motif as $v$. We consider multiple motif size and connection type choices during architectural search using continuous-valued parameterizations $\alpha^v$ and $\alpha_{ir}^c$, respectively for each motif size $v$ and connection type $c$. Instead of applying a single choice to each architectural parameter, the categorical choice of each type or size is relaxed to a softmax over all possible options which can be expressed as

$$\hat{\alpha}_{ir}^c = \frac{exp(\alpha_{ir}^c)}{\sum_{c' \in \mathcal{C}} exp(\alpha_{ir}^{c'})}, \quad \hat{\alpha}^v = \frac{exp(\alpha^v)}{\sum_{v' \in \mathcal{V}} exp(\alpha^{v'})}. \tag{13}$$

$\mathcal{C}$ and $\mathcal{V}$ are the set of all possible connection types and motif sizes, respectively; $\hat{\alpha}^v$ and $\hat{\alpha}_{ir}^c$ are the continuous-valued categorical choice of motif size $v$ and connection type $c$, respectively, which can also be interpreted as the probability of selecting the corresponding motif size or connection type. In this paper, we use hat over the variable to denote the architectural parameter processed by softmax. Then, the task of architecture optimization is reduced to learn a set of continuous variables $\hat{\alpha} = \{\hat{\alpha}_{ir}^c, \hat{\alpha}^v\}$. With the continuous architectural parameters, a gradient-based method like BP is applicable to learn the recurrent connectivity.

In Liu et al. (2018), the bi-level optimization problem is simply approximated to a one-shot model to reduce the expensive computational cost of the inner optimization which can be expressed as

$$\nabla_{\hat{\alpha}} \mathcal{L}_{valid}(\hat{\alpha}, w^*(\hat{\alpha})) = \nabla_{\hat{\alpha}} \mathcal{L}_{valid}(\hat{\alpha}, w - \eta \nabla_w \mathcal{L}_{train}(w, \hat{\alpha})), \tag{14}$$

where $\eta$ is the learning rate for a step of inner loop. Both the weights of the search network and the architectural parameters are trained by the BP method.

The architectural gradient can be approximated by

$$\frac{d\mathcal{L}_{valid}}{d\hat{\alpha}}(\hat{\alpha}) = \nabla_{\hat{\alpha}} \mathcal{L}_{valid}(\hat{\alpha}, w^*) - \eta \nabla_w \mathcal{L}_{valid}(\hat{\alpha}, w^*) \nabla_{\hat{\alpha}, w}^2 \mathcal{L}_{train}(w^*, \hat{\alpha}). \tag{15}$$

The complexity is further reduced by using the finite difference approximation around $w^{\pm} = w \pm \epsilon \nabla_w \mathcal{L}_{valid}(\hat{\alpha}, w^*)$ for small perturbation $\epsilon$ to compute the gradient of $\nabla_{\hat{\alpha}} \mathcal{L}_{valid}(\hat{\alpha}, w^*)$. Finally the architectural updates in (15) can be calculated as

$$\frac{d\mathcal{L}_{valid}}{d\hat{\alpha}}(\hat{\alpha}) = \nabla_{\hat{\alpha}} \mathcal{L}_{valid}(\hat{\alpha}, w^*) - \frac{\eta}{2\epsilon}(\nabla_{\hat{\alpha}} \mathcal{L}_{train}(w^+, \hat{\alpha}) - \nabla_{\hat{\alpha}} \mathcal{L}_{train}(w^-, \hat{\alpha})). \tag{16}$$

## C  BACKPROPAGATION VIA HRMAS FRAMEWORK

Without loss of generality, we consider a multi-layered RSNN consisting of one or more SC-ML layers, where connections between layers are assumed to be feedforward. We focus on a proposed SC-ML of an RSNN. Based on the leaky integrate-and-fire (LIF) neuron model in (7), the neuronal membrane voltage $u_i[t]$ of neuron $i$ in the SC-ML layer at time $t$ is given by integrating currents from all inter-layer inputs and intra-layer recurrent connections under all possible architectural parameterizations:

$$u_i[t] = (1 - \frac{1}{\tau})u_i[t-1] + \frac{R}{\tau}(\sum_j w_{ij} a_j[t] + \sum_{v \in \mathcal{V}} (\hat{\alpha}^v \sum_r^{I_i^v} \sum_{c \in \mathcal{C}} (\hat{\alpha}_{ir}^c w_{ir}^c a_r[t-1]))), \tag{17}$$

where $R$ and $\tau$ are the resistance and time constant of the membrane, $w_{ij}$ the synaptic weight from neuron $j$ in the previous layer to neuron $i$, $w_{ir}^c$ the recurrent weight from neuron $r$ to neuron $i$ of connection type $c$, and $a_j[t]$ the (unweighted) postsynaptic current (PSC) converted from spikes of neuron $j$ through a synaptic model. To reduce clutter in the notation, we use $I_i^v$ to denote the number of presynaptic connections afferent onto neuron $i$'s input in the recurrent layer when choosing motif size $v$, which includes both inter and intra-motif connections. We further drop the explicit dependence of $\hat{\alpha}_{ir}^c$ on $\hat{\alpha}^v$. We assume feedforward connections have no time delay and recurrent

connections have one time step delay. The response of neuron $i$ obtained from recurrent connections is the summation of all the weighted recurrent inputs over the probabilities of connection types and motif sizes.

During the learning, We define the loss function as

$$L = \sum_{k=0}^{T} E[t_k], \tag{18}$$

where $T$ is the total time steps and $E[t_k]$ the loss at $t_k$. From (17) and (8), the membrane potential $u_i[t]$ of the neuron $i$ at time $t$ demonstrates contribution to all future fires and losses of the neuron through its PSC $a_i[t]$. Therefore, the error gradient with respect to the presynaptic weight $w_{ij}$ from neuron $j$ to neuron $i$ can be defined as

$$\begin{aligned}
\frac{\partial L}{\partial w_{ij}} &= \sum_{k=0}^{T} \frac{\partial E[t_k]}{\partial w_{ij}} = \sum_{k=0}^{T} \sum_{m=0}^{k} \frac{\partial E[t_k]}{\partial u_i[t_m]} \frac{\partial u_i[t_m]}{\partial w_{ij}} \\
&= \sum_{m=0}^{T} \frac{R}{\tau} a_j[t_m] \sum_{k=m}^{T} \frac{\partial E[t_k]}{\partial u_i[t_m]} = \sum_{m=0}^{T} \frac{R}{\tau} a_j[t_m] \delta_i[t_m],
\end{aligned} \tag{19}$$

where $\delta_i[t_m]$ denotes the error for neuron $i$ at time $t_m$ and is defined as:

$$\delta_i[t_m] = \sum_{k=m}^{T} \frac{\partial E[t_k]}{\partial u_i[t_m]} = \sum_{k=m}^{T} \frac{\partial E[t_k]}{\partial a_i[t_k]} \frac{\partial a_i[t_k]}{\partial u_i[t_m]}. \tag{20}$$

In this work, the output layer is regular feedforward layer without recurrent connection. Therefore, the weight $w_{oj}$ of output neuron $o$ is updated by

$$\frac{\partial L}{\partial w_{oj}} = \sum_{m=0}^{T} \frac{R}{\tau} a_j[t_m] \sum_{k=m}^{T} \frac{\partial E[t_k]}{\partial a_o[t_k]} \frac{\partial a_o[t_k]}{\partial u_o[t_m]}, \tag{21}$$

where $\frac{\partial E[t_k]}{\partial a_o[t_k]}$ depends on the choice of the loss function.

Now, we focus on the backpropagation in the recurrent hidden layer while the feedforward hidden layer case can be derived similarly. For a neuron $i$ in SC-ML, in addition to the error signals from the next layer, the error backpropagated from the recurrent connections should also be taken into consideration. The backpropagated error can be calculated by:

$$\begin{aligned}
\delta_i[t_m] &= \sum_{k=m}^{T} \sum_{j=k}^{T} \frac{\partial a_i[t_k]}{\partial u_i[t_m]} \sum_{p=1}^{N_p} \left( \frac{\partial u_p[t_k]}{\partial a_i[t_k]} \frac{\partial E[t_j]}{\partial u_p[t_k]} \right) \\
&+ \sum_{k=m}^{T} \sum_{j=k+1}^{T} \frac{\partial a_i[t_k]}{\partial u_i[t_m]} \sum_{r}^{N_r} \left( \frac{\partial u_r[t_k+1]}{\partial a_i[t_k]} \frac{\partial E[t_j]}{\partial u_r[t_k+1]} \right) \\
&= \sum_{k=m}^{T} \frac{\partial a_i[t_k]}{\partial u_i[t_m]} \sum_{p=1}^{N} (\frac{R}{\tau} w_{pi} \delta_p[t_k]) \\
&+ \sum_{k=m}^{T-1} \frac{\partial a_i^{(l)}[t_k]}{\partial u_i^{(l)}[t_m]} \sum_{v \in \mathcal{V}} (\hat{\alpha}^v \sum_{r}^{O_i^v} \sum_{c \in \mathcal{C}} \frac{R}{\tau} \hat{\alpha}_{ri}^c w_{ri}^c \delta_r[t_k+1]),
\end{aligned} \tag{22}$$

where $N_p$ and $N_r$ are the number of neurons in the next layer and the number of neurons in this recurrent layer, respectively. $\delta_p$ and $\delta_r$ are the errors of the neuron $p$ in the next layer and the error from the neuron $r$ through the recurrent connection. $O_i^v$ represents all the postsynaptic neurons of neuron $i$'s outputs in the recurrent layer when choosing motif size $v$, which includes both inter and intra-motif connections.

The key term in (22) is $\frac{\partial a[t_k]}{\partial u[t_m]}$ which reflects the effect of neuron's membrane potential on its output PSC. Due to the non-differentiable spiking events, it becomes the main difficulty for the BP of SNNs.

Various approaches are proposed to handle this problem such as probability density function of spike state change (Shrestha & Orchard, 2018), surrogate gradient (Neftci et al., 2019), and Temporal Spike Sequence Learning via Backpropagation (TSSL-BP) (Zhang & Li, 2020b).

With the error backpropagated according to (22), the weights and architectural parameters can be updated by gradient descent as:

$$\Delta w_{ij} \propto \delta_i[t]\frac{R}{\tau}a_j[t], \quad \Delta\hat{\alpha}^v \propto \sum_i^{N_r} \delta_i[t]\frac{R}{\tau}\sum_r^{I_i^v}(\sum_{c\in\mathcal{C}}\hat{\alpha}_{ir}^c w_{ir}^c a_r[t-1]),$$
$$\Delta w_{ir}^c \propto \delta_i[t]\frac{R}{\tau}\sum_{v\in\mathcal{V}}(\hat{\alpha}^v\hat{\alpha}_{ir}^c a_r[t-1]), \quad \Delta\hat{\alpha}_{ir}^c \propto \delta_i[t]\frac{R}{\tau}\sum_{v\in\mathcal{V}}(\hat{\alpha}^v w_{ir}^c a_r[t-1]). \tag{23}$$

## D  SPIKL-IP

In this work, we apply the SpiKL-IP (Zhang & Li, 2019a) to all the recurrent neurons which not only maintains the network activity but also maximizes the information of neurons' outputs from an information-theoretic perspective. More specifically, SpiKL-IP adapts the intrinsic parameters of a spiking neuron while minimizing the KL-divergence from the targeted exponential distribution to the actual output firing rate distribution. The two neuronal parameters $R$ and $\tau$ are online updated according to the approximate average firing rate of the neuron by:

$$\Delta R = \frac{2y\tau V_{th} - W - V_{th} - \frac{1}{\mu}\tau V_{th}y^2}{RW}, \quad \Delta\tau = \frac{-1+\frac{y}{\mu}}{\tau}, \quad W = \frac{V_{th}}{e^{\frac{1}{\tau y}}-1}, \tag{24}$$

where $\mu$ is the desired mean firing rate, $y$ the average firing rate of the neuron. Similar to biological neurons, we use the intracellular calcium concentration $\phi[t]$ as a good indicator of the averaged firing activity and y can be expressed with the time constant of calcium concentration $\tau_{cal}$ as

$$\phi_i[t] = (1-\frac{1}{\tau_{cal}})\phi_i[t-1] + s_i[t], \quad y_i[t] = \frac{\phi_i[t]}{\tau_{cal}}. \tag{25}$$

We explicitly express the neuronal parameters $R$ and $\tau$ of neuron $i$ tuned through time as $R_i[t]$ and $\tau_i[t]$, since they are adjusted by the IP rule at each time step. They are updated by

$$R_i[t] = R_i[t-1] - \gamma\Delta R_i, \quad \tau_i[t] = \tau_i[t-1] - \gamma\Delta\tau_i, \tag{26}$$

where $\gamma$ is the learning rate of the SpiKL-IP rule.

In addition, by including time-variant neuronal parameters $R$ and $\tau$ into (22) and (23), the one time step architectural parameter and weight updates change to

$$\delta_i[t_m] = \sum_{k=m}^{T}\frac{\partial a_i[t_k]}{\partial u_i[t_m]}\sum_{p=1}^{N}(\frac{R_p[t_k]}{\tau_p[t_k]}w_{pi}\delta_p[t_k])$$
$$+ \sum_{k=m}^{T-1}\frac{\partial a_i^{(l)}[t_k]}{\partial u_i^{(l)}[t_m]}\sum_{v\in\mathcal{V}}(\hat{\alpha}^v\sum_r^{O_i^v}\sum_{c\in\mathcal{C}}\frac{R_r[t_k+1]}{\tau_r[t_k+1]}\hat{\alpha}_{ri}^c w_{ri}^c\delta_r[t_k+1])$$
$$\Delta w_{ij} \propto \delta_i[t]\frac{R_i[t]}{\tau_i[t]}a_j[t], \quad \Delta\hat{\alpha}^v \propto \sum_i^{N_r}\delta_i[t]\frac{R_i[t]}{\tau_i[t]}\sum_r^{I_i^v}(\sum_{c\in\mathcal{C}}\hat{\alpha}_{ir}^c w_{ir}^c a_r[t-1])$$
$$\Delta w_{ir}^c \propto \delta_i[t]\frac{R_i[t]}{\tau_i[t]}\sum_{v\in\mathcal{V}}(\hat{\alpha}^v\hat{\alpha}_{ir}^c a_r[t-1]), \quad \Delta\hat{\alpha}_{ir} \propto \delta_i[t]\frac{R_i[t]}{\tau_i[t]}\sum_{v\in\mathcal{V}}(\hat{\alpha}^v w_{ir}^c a_r[t-1]), \tag{27}$$

## E  EXPERIMENTS

The proposed HRMAS framework with SC-ML is evaluated on speech dataset TI46-Alpha (Liberman et al., 1991), neuromorphic speech dataset N-TIDIGITS (Anumula et al., 2018), neuromorphic

video dataset DVS-Gesture (Amir et al., 2017), and neuromorphic image dataset N-MNIST (Orchard et al., 2015). The performances are compared with several existing results on different structures of SNNs and ANNs such as feedforward SNNs, RSNNs, Liquid State Machine(LSM), LSTM, and so on. We will share our Pytorch (Paszke et al., 2019) implementation on GitHub. We expect this work would motivate the exploration of RSNNs architecture in the neuromorphic community.

### E.1    EXPERIMENTAL SETTINGS

All reported experiments are conducted on an NVIDIA RTX 3090 GPU. The implementation of the proposed methods is on the Pytorch framework (Paszke et al., 2019).

In the SNNs of the experiments, the fully connected weights between layers are initialized by the He Normal initialization proposed in He et al. (2015). The recurrent weights of excitatory connections are initialized to $0.2$ and tuned by the BP method. The weights of inhibitory connections are initialized to $-2$ and fixed. The simulation step size is set to 1 ms. The parameters like thresholds and learning rate are empirically tuned. No synaptic delay is applied for feedforward connections while recurrent connections have 1 time step delay. No refractory period, normalization, or dropout is used. Adam (Kingma & Ba, 2014) is adopted as the optimizer. The mean and standard deviation (std) of the accuracy reported is obtained by repeating the experiments five times.

Our experiments contain two phases. In the first phase, the weights are trained via the training set while the validation set is used to optimize architectural parameters. In the second phase, the motif topology and type of lateral connections are fixed after obtaining the optimal architecture. All the weights of the network are reinitialized. Then, the new network is trained on the training set and tested on the testing set. The test performance is reported in the paper. In addition, since all the datasets adopted in this paper only contain training sets and testing sets, our strategy is to divide the training set. In the first phase, the training set is equally divided into a training subset and a validation subset. Then, the architecture is optimized on these subsets. In the second phase, since all the weights are reinitialized, we can train the weights with the full training set and test on the testing set. Note that the testing set is only used for the final evaluation.

Table 3 lists the typical constant values of parameters adopted in our experiments for each dataset. The SC-ML size denotes the number of neurons in the SC-ML. In our experiments, each network contains one SC-ML as the hidden layer. In addition, five motif sizes are predetermined before the experiment. The HRMAS framework optimizes the motif size from one of the five options.

| Parameter | TI46-Alpha | N-TIDIGITS | DvsGesture | N-MNIST |
|:---:|:---:|:---:|:---:|:---:|
| $\tau_m$ | 16 ms | 64 ms | 64 ms | 16 ms |
| $\tau_s$ | 8 ms | 8 ms | 8 ms | 8 ms |
| $\tau_{cal}$ | 16 ms | 16 ms | 16 ms | 16 ms |
| learning rate | 0.0005 | 0.0005 | 0.0001 | 0.0005 |
| Batch Size | 50 | 50 | 20 | 50 |
| Time steps | 100 | 300 | 400 | 100 |
| Epochs for searching | 300 | 200 | 60 | 30 |
| Epochs for testing | 400 | 400 | 150 | 100 |
| SC-ML size | 800 | 800 | 512 | 512 |
| Motif size options | [5, 10, 16, 25, 40] | | [2, 4, 8, 16, 32] | |

Table 3: Parameters settings.

### E.2    LOSS FUNCTION

For the BP method used in this work, the loss function can be defined by any errors that measure the distance between the actual outputs and the desired outputs. In our experiments, since hundreds of time steps are required for simulating speech and neuromorphic inputs, we choose the accumulated output PSCs to define the error which is similar to the firing count used in many existing works (Jin et al., 2018; Shrestha & Orchard, 2018).

We suppose the simulation time steps for a sample is $T$. In addition, for neuron $o$ of the output layer, we define the desired output as $d_o$ and real output as $r_o$ where $r_o = \sum_{k=1}^{T} a_o[t_k]$ and $d_o$ is manually determined. Therefore, the loss is determined by the square error of the outputs

$$L = \sum_{k=1}^{T} E[t_k] = \sum_{o}^{N^{(out)}} \frac{1}{2}(d_o - r_o)^2, \tag{28}$$

where $N^{(out)}$ is the number of neurons in the output layer.

Furthermore, the error at each time step is simply defined by the averaged loss through all the time steps:

$$E[t_k] = \frac{L}{T}, \quad E_o[t_k] = \frac{(d_o - r_o)^2}{2T}. \tag{29}$$

With the loss function defined above, the error $\delta$ can be calculated for each layer according to (22).

### E.3 Datasets

TI46 speech corpus (Liberman et al., 1991) contains spoken English alphabets and digits audios from 16 speakers. In our experiments, the full alphabets subset of the TI46 is used and dubbed TI46-Alpha. The TI46-Alpha has 4142 and 6628 spoken English examples in 26 classes for training and testing, respectively. The continuous temporal speech waveforms are preprocessed by Lyon's ear model (Lyon, 1982) which is the same as the preprocessing steps in Zhang & Li (2019b). The sample rate of this dataset is 12.5 kHz. The decimation factor of Lyon's ear model is 125. The MATLAB implementation of Lyon's ear model is available online (Slaney, 1998). Each sample is encoded into 78 channels. In our experiments, the preprocessed real-value intensities are directly applied as the inputs.

The N-TIDIGITS (Anumula et al., 2018) is the neuromorphic version of the speech dataset Tidigits (Leonard & Doddington, 1993). The original audios are processed by a 64-channel CochleaAMS1b sensor and recorded as the spike responses. The dataset contains both single-digit samples and connected-digit sequences with a vocabulary consisting of 11 digits including "oh," "zero" and the digits "1" to "9". In the experiments, only single-digit samples are used. In total, there are 55 male and 56 female speakers with 2475 single-digit samples for training and the same number of samples for testing. In the original dataset, each sample has 64 input channels and takes about $0.9\ s$. To speed up the simulation, each sample is reduced to 300 time steps by compressing the time resolution from $1\ us$ to $3\ ms$. During the compression, a channel has a spike at a certain time step in the preprocessed sample if it contains at least one spike in the corresponding time window of the original sample.

The DVS-Gesture dataset (Amir et al., 2017) consists of recordings of 29 different individuals (subjects) performing hand and arm gestures. The spikes are generated from natural motion. There are 122 trials in total. Each trial contains the recording for one subject by a dynamic vision sensor (DVS) camera under one of the three different lighting conditions. In each trial, 11 hand and arm gestures of the subject are recorded. Samples from the first 23 subjects are used for training and the other 6 subjects for testing. During preprocessing, the trials are separated into individual actions (gestures). The task is to classify the action sequence video into an action label. Each action (sample) lasts for about $6\ s$. In addition, two channels with $128 \times 128$ pixels in each channel are recorded. We compress the temporal resolution to $15\ ms$ which means it takes 400 time steps for each sample. Similar to the preprocessing of N-TIDIGITS, the input pixel has a spike at a certain time step in the preprocessed sample if it contains at least one spike in the corresponding $15\ ms$ time window of the original sample. In the experiments, the inputs are first processed by the pooling layer of $4 \times 4$ pooling kernel size. Thus, the inputs to the hidden layer have 2 channels with the size of $32 \times 32$ in each channel.

The N-MNIST dataset (Orchard et al., 2015) is a neuromorphic version of the MNIST dataset generated by tilting a DVS in front of static digit images on a computer monitor. The movements inducing pixel intensity changes at each location are encoded as spike trains. Since the intensity can either increase or decrease, two kinds of ON- and OFF-events spike events are recorded. Due to the relative shifts of each image, an image size of $34 \times 34$ is produced. Each sample of the N-MNIST is a spatio-temporal pattern with $34 \times 34 \times 2$ spike sequences lasting for $300ms$ with the resolution of

$1us$. In our experiments, we reduce the time resolution of the N-MNIST samples by 3000 times to speed up the simulation. Therefore, the preprocessed samples only have about 100 time steps.

# F ADDITIONAL INFORMATION

## F.1 ADDITIONAL DATA

We provided additional experimental data in Table 4, including: random search in the search space as the baseline for HRMAS, effect of IP rule, HRMAS perference on Larger network. Experimental results show that: the HRMAS method shows consistent superiority (around 2%) over the random search baseline; IP rule brings stable performance improvement (around 1.3%); our method can be efficiently extended to networks with more neurons while providing good performance.

Table 4: Test Accuracy on TI46-Alpha, obtained by repeating 5 times with different random seeds, including: HRMAS perference on Larger network, effect of IP rule, random search as the baseline architecture.

| Arch Optimization | Learning Rule | SC-ML Sizes | Best | Mean | Std |
|---|---|---|---|---|---|
| HRMAS (with IP) | TSSL-BP | 800 | **96.44%** | **96.08%** | **0.27%** |
| HRMAS (with IP) | TSSL-BP | 1600 | **96.26%** | - | - |
| HRMAS (with IP) | TSSL-BP | 2400 | **96.58%** | - | - |
| HRMAS (with IP) | TSSL-BP | 3200 | **96.45%** | - | - |
| HRMAS (w/o IP) | TSSL-BP | 800 | 95.17% | 94.74% | 0.32% |
| Random | TSSL-BP | 800 | 94.47% | 94.18% | 0.30% |

## F.2 THE COMPUTATIONAL RESOURCES REQUIRED FOR OUR METHOD

The proposed HRMAS bi-level optimization process is similar to DARTS, so the overall computational complexity is similar to DARTS; the IP method is an localized unsupervised learning method and does not constitute significant computational consumption. Furthermore, our proposed SC-ML topology greatly reduces the search space. Specifically, as the Figure 9 shows, the HRMAS optimization of a SC-ML layer, with $n$ neurons, a motif size of $s$ and $n/s$ motifs, reduces the parameters that need to be optimized for the recurrent connection matrix from $O(n^2)$ to $O(sn)$: $O(n/s)$ inter-motif connections + $O(n/s * s^2)$ intra-motif connections + $O(n)$ neuron hyperparameters. Generally, $s \ll n$, which reduces the parameter space of recurrent connections to linear growth with the neuron numbers, allowing our algorithm scale well. Specifically, a complete training process of a RSNN with 800 neurons hidden layer for TI46-alpha dataset, including 150 epoch for cell size search, 150 epoch for connection type search and 400 epoch for finetune, takes 4 hours on single NVIDIA GeForce RTX 3090 GPU.

## F.3 IP RULE'S EFFECT ON PERFORMANCE DURING OPTIMIZING PROCESS

We plotted the performance curve of the network optimization process on the TI46-alpha dataset. Figure 7 and Figure 8 show the loss and accuracy on the validation set respectively. The solid line and shading show the mean and standard deviation of the 5 experiments. We conducted experiments with ip rule turned on and off. We use green text to mark each phase of architecture optimization.

The experimental results show: 1.When the network architecture changes drastically, such as iteration = 750 (the cell size search ends and the connection type search starts), and iteration = 1500 (the connection type search ends, the network is discretized and fine-tuned), the network There will be a slight performance degradation. But it can be quickly improved to a higher level by the next stage of training. 2.It can be found that IP method brings two benefits: improved network performance and a more stable training process. The figures showed that the red solid line (mean) has always performed better than the blue solid line without the IP method; at the same time, the red shadow (standard deviation) has always been narrower than the blue shadow, which means a more stable network architecture search process.

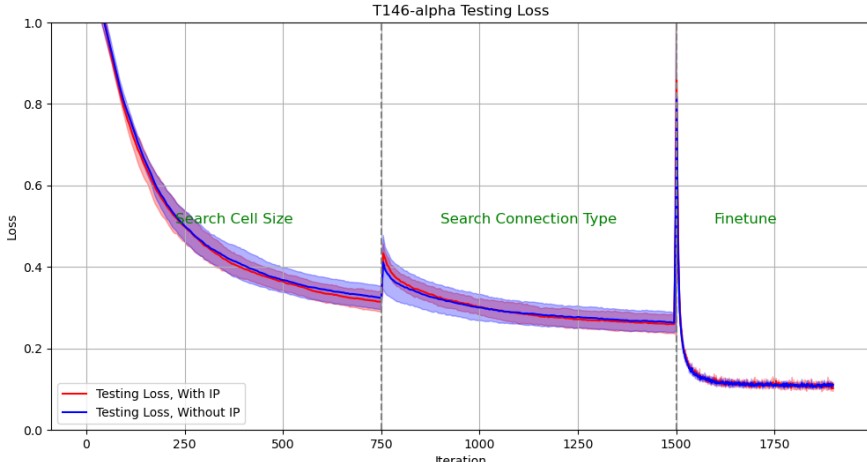

Figure 7: Test Loss in architectural optimization in HRMAS. The solid line and shading show the mean and standard deviation of the 5 experiments. We conducted experiments with ip rule turned on (red) and off (blue). It can be found that IP method brings two benefits: improved network performance and a more stable training process. The figures showed that the red solid line (mean) has lower loss than the blue solid line without the IP method; at the same time, the red shadow (standard deviation) has always been narrower than the blue shadow, which means a more stable network architecture search process.

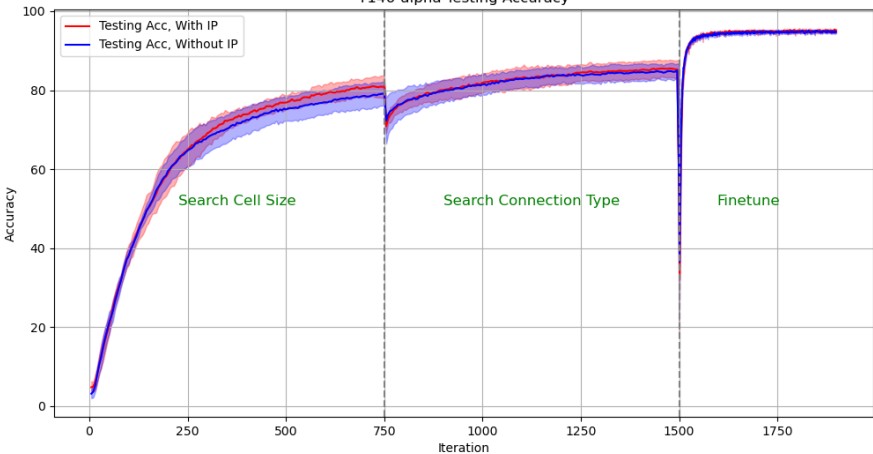

Figure 8: Test Accuracy in architectural optimization in HRMAS. The solid line and shading show the mean and standard deviation of the 5 experiments. We conducted experiments with ip rule turned on (red) and off (blue). It can be found that IP method brings two benefits: improved network performance and a more stable training process. The figures showed that the red solid line (mean) has higher accuracy than the blue solid line without the IP method; at the same time, the red shadow (standard deviation) has always been narrower than the blue shadow, which means a more stable network architecture search process.

## F.4 SPARSITY RECURRENT CONNECTION OF OPTIMIZED RSNN

We shows in the Figure 9 the weight matrix of the RSNN with SC-ML optimized by the HRMAS method. The original fully connected recurrent matrix size is 800*800. We set the search space of motif size to [2,4,8,16,32]. In five random experiments, the HRMAS optimization method always gave the search results of motif-size=2, with similar inter/intra motif topology. This limits the huge recurrent matrix to a highly sparse band matrix with non-zero values only near the diagonal, greatly reducing the search space, parameter amount, and optimization difficulty.

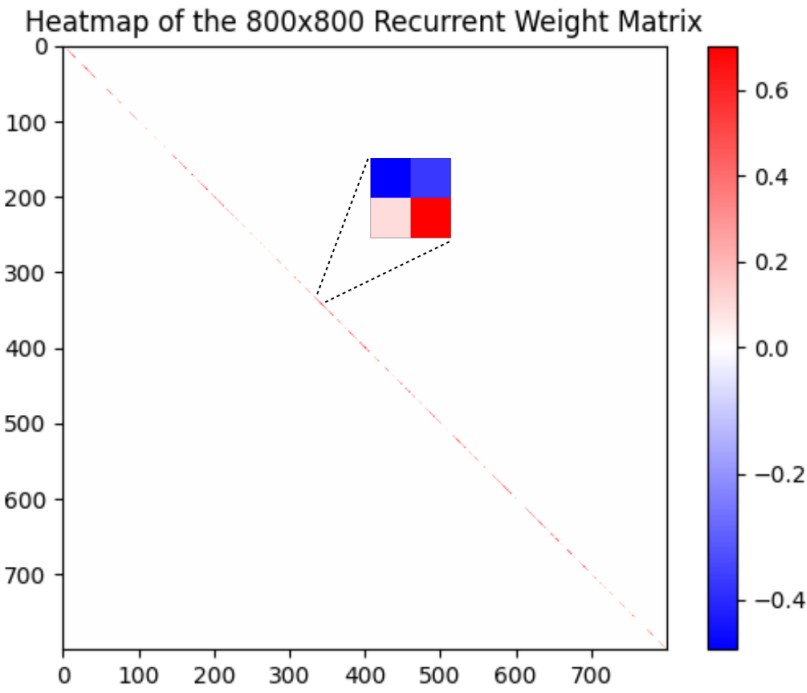

Figure 9: Recurrent Weight Matrix after optimization by HRMAS. The original fully connected recurrent matrix size is 800*800. We set the search space of motif size to [2,4,8,16,32]. In five random experiments, the HRMAS optimization method always gave the search results of motif-size=2, with similar inter/intra motif topology. This limits the huge recurrent matrix to a highly sparse band matrix with non-zero values only near the diagonal, greatly reducing the search space, parameter amount, and optimization difficulty.

