# OpenReview forum: "Composing Recurrent Spiking Neural Networks using Locally-Recurrent Motifs and Risk-Mitigating Architectural Optimization"
_ICLR.cc/2024/Conference — Submitted to ICLR 2024_

### Official Review · Reviewer_jEHG · 2023-10-14

**Soundness:** 3 good
**Presentation:** 4 excellent
**Contribution:** 2 fair
**Rating:** 5
**Confidence:** 4

**Summary:**

The authors propose a novel bi-level optimization method for architecture search in recurrent spiking neural network models. The authors compose RSNNs using a sparsely connected recurrent Motif Layer (SC-ML), which consists of multiple small recurrent motifs wired together by sparse lateral connections. They also propose a method called Hybrid Risk-Mitigating Architectural Search (HRMAS) to systematically optimize the topology of the proposed recurrent motifs and SC-ML layer architecture.

**Strengths:**

The notion of a systematic architecture search process for RSNNs is a very interesting research question.
The authors showed the performance of their method on some standard tasks is comparable or better than current methods, which is good.
The notion of using intrinsic plasticity for an unsupervised self-repairing model was very interesting

**Weaknesses:**

Though the paper showed promising results, some major issues need to be addressed in the paper:

The major contributions of this paper seem to be just taking the current DARTS-based architecture search method used in DNNs and using it for RSNNs.  I would recommend the authors to highlight their key contributions ( see Questions for details)

The paper introduces two different things, and as such, the experimental section is extremely weak and does not give sufficient evidence of the model performance and how the introduced concepts help in designing a better RSNN model. The absence of more complete ablation studies and specific case scenarios limits the understanding of the necessity and impact of certain methodological choices.

The paper does not delve into the computational complexity and practicality of the proposed methods, especially in terms of time and resources required for simulations and optimizations.

The robustness and stability of the proposed methods, especially in the context of different initializations and model sizes, are not thoroughly validated.

**Questions:**

A. SPARSELY-CONNECTED RECURRENT MOTIF LAYER (SC-ML)

1. Since the authors use the same topology for all the motifs, it would be good if the authors could elucidate how they chose this topology and what effect it has on the final architecture. I feel this notion of motifs is inspired by the blocks building the cells in the DARTS paper - if so, the DARTS method used a variety of different convolution layers for these blocks. It would be interesting if the authors could highlight why no such heterogeneity would be required in designing RSNNs.

2. The SC-ML architecture seems very similar to the concept of clustered ESNs. Can you give some explanation of how this is similar/different?

B. HYBRID RISK-MITIGATING ARCHITECTURAL SEARCH (HRMAS)

1. The authors introduce the HRMAS as a bi-level optimization where, in the first step, they optimize $\alpha$ and $w$ hierarchically, based on gradient-based optimization. In the second step, they use IP to adapt the parameters of each neuron over a time window. However, that does not match with the problem formulation in Eqs 1-3, where it seems the first-level optimization searches for the architecture, the second-level searches for the optimal  parameters, and the third level optimizes the weights

2. The authors repeatedly mention the “risks” caused by the change in the first level of the bi-level optimization. It would be good if the authors were more specific and showed a more complete ablation study of what happens if they do not use this IP optimization.

3. It seems the gradient-based optimization is simply the DARTS method - it would be good if the authors could highlight the novelty of the architecture search method and how they are optimized for spiking neural networks and the neuronal timescales (as they mentioned in the abstract and introduction) Right now, it seems the architecture search is an off the shelf implementation of the DARTS bi-level optimization problem.

4. Since this is an architecture search process, it would be recommended that the authors also add the following results:
       a. performance comparison with random weights/architecture (this should be the baseline)
       b. the complexity of the algorithm - like how long it took to run these simulations and how the performance changes over the iterations of the optimization problem

5. The authors propose this method can be used to design very large RSNN models. The results in Table 1 show the size is comparable to current RSNN models. It would be interesting if the authors could give more results on what happens if the number of neurons increases.

6. From Figure 6, it seems the final model is a densely connected model compared to many of the current methods. As mentioned before, an important ablation study would be to compare with a randomly generated model (normally these models are much sparsely connected). If there is a significant difference, it would be interesting to know why such dense connections are important for better performance

7. From my personal experiences, such architecture search models are extremely unstable and highly dependent on the initializations. Can you add some details on the initialization you used for your experiments and whether two different initialization models converge to similar or very different models?  It is also important that the authors rerun the experiments a few times (with same/different initializations) and report the mean and variance of the performance in Table 1.

---

> ### Author Response · Authors · 2023-11-20
> **Thanks for your insightful questions.**
>
> QA1: We admit that the concept of motif is similar to the cell in DARTs as the most basic computing unit. It is worth noting that both our SC-ML and HRMAS methods can be seamlessly extended to RSNN with different motif sizes. In the paper, we mainly consider computational complexity: heterogeneous motifs will bring significantly larger search spaces and reduce the efficiency of the NAS algorithm. By limiting motifs to the same size, we achieve a balance between search space size, network topology sparsity, and performance.
>
> QA2: Our network does share topological similarities with ESN: there are recurrent connections within a single neuron layer, which complicates the spatiotemporal dynamic nature of the neuron layer and increases its information processing capabilities. However, the weights of most ESN networks are randomly set and not optimized, which significantly limits the learning capabilities of these networks. Our method can make all weight parameters in the network learnable by designing the SC-ML topology and HRMAS optimization method. This significantly improves the perfermence and applicability of RSNN.
>
> QB1: We would like to point out that eq1-3 are only the complete theoretical form of the entire optimization problem, instead of the specific optimization process. Our bi-level optimization method is an approximate solution of this problem to alleviate the computational overhead.
>
> QB2 & QB4 & QB5 & QB6: Please check general response/Section: Additional Information in Appendix of our paper. We provide comprehensive data & analysis of learned recurrent matrix, computational resources, random search baseline, learning curve and ablation study of IP rule there.
>
> QB3: We acknowledge that our current approach is based on DARTS. Our novelty lies in: we design the SC-ML layer to reduce the size of the network search space while ensuring the sparsity of recurrent connections. At the same time, in order to optimize this topology, we designed the HRMAS algorithm including IP to stabilize the network
> function and mitigate potential risks caused by architectural changes.
>
> QB7: We rerun the experiment 5 times using different random seeds for initialization. The results are as follows. It can be seen that our optimization algorithm is not sensitive to the choice of initialization and can converge stably. We also provided more data in general response, more figures of complete learning curve with different initializations in Section: Additional Information in Appendix of our paper.
>
> | Dataset     | Learning Rule | Hidden Layers | Mean      | Std      | Best        |
> |-------------|---------------|---------------|-----------|----------|-------------|
> | TI46-Alpha  | TSSL-BP       | 800           | 96.08%    | 0.27%    | **96.44%**  |
> | N-TIDIGITS  | TSSL-BP       | 400           | 94.27%    | 0.35%    | **94.66%**  |
> | DVS-Gesture | TSSL-BP       | P4-512        | 88.40%    | 1.71%    | **90.28%**  |
> | N-MNIST     | TSSL-BP       | 512           | 98.60%    | 0.08%    | **98.72%**  |

---

### Official Review · Reviewer_8BVm · 2023-10-31

**Soundness:** 2 fair
**Presentation:** 2 fair
**Contribution:** 2 fair
**Rating:** 5
**Confidence:** 4

**Summary:**

To solve the systemic architectural optimization of RSNNs, this paper proposed RSNN models with scalable architecture and automated architectural optimization. The proposed RSNN is composed based on a layer architecture called Sparsely-Connected Recurrent Motif Layer (SC-ML) that consists of multiple small recurrent motifs wired together by sparse lateral connections. They claim that the small size of the motifs and sparse inter-motif connectivity leads to an RSNN architecture scalable to large network sizes. The Hybrid Risk-Mitigating Architectural Search (HRMAS) is designed to systematically
optimize the topology of the proposed recurrent motifs and SC-ML layer architecture. The experiments are conducted on one speech and three neuromorphic datasets, and the results demonstrate the performance improvement brought by the proposed automated architecture optimization over existing manually-designed RSNNs.

**Strengths:**

The architecture optimization in RSNNs is important. The problem that this paper tries to solved is interesting.

**Weaknesses:**

* About the presentation

The figure 4 is not clear to show the mechanism of the HRMAS. More explanation can be added.
Also the figure 2 is not clear enough to show the architecture optimization in HRMAS.

* About the performance

Although there is little accuracy improvements brought by the proposed method. But the accuracy increasing is quite limited in these datasets, especially on the N_TIDIGITS and DVS-GESTURE and N_MNIST.

* About the experimental results except the accuracy

Since the architecture optimization proposed in this paper is complex, how about the training speed and computation resources consumption? Only accuracy comparison seems not sufficient for ICLR publication.

**Questions:**

1. Please refer to the above weakness.
2. If the proposed RSNN model is applied to larger datasets such as DVS-CIFAR10, how about the training speed and computation resources consumption?

**Details Of Ethics Concerns:**

None.

---

> ### Author Response · Authors · 2023-11-20
> **Thanks for your insightful questions.**
>
> Q1: The figure 4 is not clear to show the mechanism of the HRMAS. More explanation can be added. Also the figure 2 is not clear enough to show the architecture optimization in HRMAS.
>
> A1: Figure 2 shows the architecture search process in HRMAS. We first determine the motif size in the supernet, and then determine the connection types of intra and inter-connection (excitatory, inhibitory and non-existent). Figure 4 shows the complete optimization process of HRMAS: horizontally, we now use BP and IP to alternately optimize the weight parameter $\omega_1$ and neuron hyperparameter $\beta$ under a certain architecture $\alpha$, and obtain the trained weight parameter $\omega_1'$, $\omega_1''$ in this process. After this process is completed, we use BP to update the network architecture parameter $\alpha$ to obtain $\alpha'$ and repeat the above optimization process under the new architecture parameter $\alpha'$.
>
> Q2: Although there is little accuracy improvements brought by the proposed method. But the accuracy increasing is quite limited in these datasets, especially on the N-TIDIGITS and DVS-GESTURE and N-MNIST.
>
> A2: We acknowledge that the progress on the dataset is not significant enough. However, the core contribution of our paper is that we are the first to propose a NAS framework that can effectively optimize the recurrent spiking neural network architecture, and prove the effectiveness of the framework through the dataset and small-scale network in the paper. Our optimization framework can be further used to build and optimize more complex RSNNs, making performance improvement an addressable problem.
>
> Q3: Since the architecture optimization proposed in this paper is complex, how about the training speed and computation resources consumption? Only accuracy comparison seems not sufficient for ICLR publication.
>
> A3: The proposed HRMAS bi-level optimization process is similar to DARTS, so the overall computational complexity is similar to DARTS; the IP method is an localized unsupervised learning method and does not constitute significant computational consumption. Furthermore, our proposed SC-ML topology greatly reduces the search space. Specifically, as the Figure9 in Appendix  shows, the HRMAS optimization of a SC-ML layer, with $n$ neurons, a motif size of $s$ and $n/s$ motifs, reduces the parameters that need to be optimized for the recurrent connection matrix from $O(n^2)$ to $O(sn)$: $O(n/s)$ inter-motif connections + $O(n/s * s^2)$ intra-motif connections + $O(n)$ neuron hyperparameters. Generally, $s \ll n$, which reduces the parameter space of recurrent connections to linear growth with the neuron numbers, allowing our algorithm scale well. Specifically, a complete training process of a RSNN with 800 neurons hidden layer for TI46-alpha dataset, including 150 epoch for cell size search, 150 epoch for connection type search and 400 epoch for finetune, takes 4 hours on single NVIDIA GeForce RTX 3090 GPU.
>
>
> Q4: If the proposed RSNN model is applied to larger datasets such as DVS-CIFAR10, how about the training speed and computation resources consumption?
>
> A4: Our proposed method is highly scalable. As the Figure 9 in Appendix shows, for a recurrent connection layer with n neurons, our method essentially reduces the optimization problem of a fully connected matrix with $O(n^2)$ parameters to $O(n)$ level. This keeps the cost of network architecture search within a controllable range when the complexity of the data set and the size of the network increase.
>
> Thanks for your insightful question! We'd be glad to discuss any unclear details further.

---

### Official Review · Reviewer_XbPq · 2023-10-31

**Soundness:** 3 good
**Presentation:** 2 fair
**Contribution:** 2 fair
**Rating:** 5
**Confidence:** 3

**Summary:**

This paper is well-composed and delves into the design and optimization of recurrent spiking neural networks (RSNNs). The authors introduce a scalable architecture called Sparsely Connected Recurrent Motif Layer (SC-ML) that uses small recurrent motifs connected sparsely. To optimize this architecture, they present the Hybrid Risk-Mitigating Architectural Search (HRMAS) method, which incorporates a biologically inspired "self-repairing" mechanism through intrinsic plasticity.

**Strengths:**

The paper is interesting as the proposed method addresses the design and optimization for the RSNN. Results from the experiments suggest that the RSNN, when integrated with SCML and Spike-IP, not only matches the performance of other SNN models but also upholds a high degree of biological accuracy. The composition of the paper is clear, making it reader-friendly and engaging.

**Weaknesses:**

- The paper lacks a comparative analysis with contemporary NAS methods for SNNs, such as “Autosnn: Towards energy-efficient spiking neural networks” and “Neural architecture search for spiking neural networks.

- The choice of baselines across different datasets lacks uniformity. The rationale behind using different baselines for each dataset remains unclear.

- The paper omits crucial metrics such as training cost, energy efficiency, latency, and hardware compatibility. While accuracy is discussed, potential strengths of the algorithm in these areas remain unexplored.

- The authors highlight the compactness of motifs and sparse connectivity between motifs as factors that make the RSNN architecture scalable via “The small size of the motifs and sparse inter-motif connectivity leads to an RSNN architecture scalable to large network sizes”. However, the significance of this claim, especially in contrast to existing state-of-the-art methods, is neither elaborated upon nor supported with empirical evidence.

- Weight evolution and architecture evolution for the algorithms are not shown. These are critical as these give crucial insight into the working of the algorithm.

**Questions:**

- Can the authors provide insights into the evolution of motif topologies during training? Such an evolution could offer valuable insights into the algorithm's adaptability and optimization process.

- There is inconsistency in the choice of baselines across datasets. The comparison of unsupervised learning rules with supervised ones seems inappropriate. As anticipated, unsupervised methods would underperform compared to their supervised counterparts.

-  The authors make claims about the Sparsely-Connected Recurrent Motif Layer (SC-ML) being able to identify sparsely connected motifs. Are there any empirical results that support this assertion?

- The algorithm hasn’t been compared to un-optimized motif networks. Such a comparison seems like a more pertinent baseline than what has been chosen for the paper.

- A comparative evaluation of the proposed method with leading NAS methods for SNNs would enhance the paper's credibility and relevance. Are there plans to include such a comparison in the future?

---

> ### Author Response · Authors · 2023-11-20
> **Thanks for your insightful questions.**
>
> Q1: Here we provide a mathematical analysis of the recurrent connection matrix of SC-ML to provide more insights into motif topologies. For the fully connected recurrent weight matrix, the number of parameters is large and difficult to optimize. In order to solve this problem, we design SC-ML's topologies to turn recurrent matrix into a Band Matrix and use HRMAS to optimize it. Specifically, intra-motif connections constitute the sub-matrices on the diagonal of the recurrent matrix; inter-motif connections constitute the values on several sub-diagonals of the recurrent matrix; motif-size determines the size of the diagonal submatrix. In the network architecture search stage: in the first step we determine the size of its diagonal submatrix (motif size), and in the second step we determine the connection type on the band (inter and intra-motif topologies), including excitability and inhibition and non-existent connections. We then used BP to optimize the specific values of the excitatory connections in the band, and used IP to optimize neuron hyperparameters to slow down performance changes caused by network architecture changes.
>
> The aforementioned configuration was designed to allow flexibility in constructing a high-performance sparse spiking RNN, while simultaneously constraining the search space to ensure practical viability in solving the complex optimization problem associated with RNN architecture. The proposed optimization framework is highly flexible and can accommodate other types of sparsity constraints.
>
>  Q2: The criterion for our selection of baselines is to align the recurrent spiking neural networks architecture. We acknowledge that the comparison with unsupervised algorithms (e.g., STDP in HeNHeS) (Chakraborty \& Mukhopadhyay, 2023) lacks fairness, but it is not the primary comparison. Except for this work, all other baselines use supervised algorithms for learning.
>
>
>  Q3 & Q4: Please check general response/Section: Additional Information in Appendix of our paper. We provide comprehensive data & analysis of learned recurrent matrix, random search baseline, learning curve and ablation study of IP rule there.
>
>
> Q5: Yes, the methods in "AutoSNN: Towards energy-efficient spiking neural networks" and "Neural architecture search for spiking neural networks" are very interesting. AutoSNN focuses on spike-aware NAS to find architectures that reduce spike count while maintaining accuracy, whereas NASSNN explores architectures with both forward and backward connections to enhance temporal information processing in SNNs. However, we have to point out that the network architectures used in these two works are both not RSNN, and the proposed methods are not directly applicable to RSNN. Although direct comparisons are not appropriate, we will explore the possibility of combining different methods in the future.
>
> Thanks for your insightful question! We'd be glad to discuss any unclear details further.

---

### Official Review · Reviewer_H38J · 2023-11-07

**Soundness:** 3 good
**Presentation:** 3 good
**Contribution:** 3 good
**Rating:** 6
**Confidence:** 4

**Summary:**

This paper developed a new neural architecture search algorithm in the recurrent spiking neural networks search space. Analogous to a cell in the NASNet search space, they defined a sparse connected motif layer (SC-ML). A recurrent spiking neural network is formed by stacking several of these SC-ML layers. The SC-ML layer has N motifs where each motif is comprised of recurrent spiking neurons of a fixed size. The neurons within a motif and between two motifs are connected using excitatory, inhibitory and non-existent connections. Further, in order to reduce the number of recurrent connections, a motif is restricted to be connected to only its neighboring motif rather than any N-1 motifs in the layer.
    In the search step, the algorithm finds the optimal motif size, the intra-motif and the inter-motif connection types. They designed a supernet which has all possible motif sizes and the intra-motif/inter-motif connection types. For intra-motif connections, a connection matrix similar to an adjacency matrix determines the kind of connection between neuron i and j.  Similar to DARTS, the motif sizes and the connection types are relaxed to form continuous probability predictions. A gradient optimization based bi-level optimization algorithm is used to perform the search, where the architecture parameters and the neural network weights are optimized alternately. In addition to that at every step, SPiKL-IP based intrinsic plasticity is used to adapt the spiking neurons to the changing network weights and the architecture weights. Upon convergence, the discretization step is performed  to obtain the best RSNN architecture.
     They evaluated the search algorithm on 3 datasets.

**Strengths:**

1. It is the first paper to perform neural architecture search for recurrent spiking neural networks. Using motifs and having intra-motif and inter-motif connections, the model's connections are no longer unwieldy and is easier to train.
2. In their ablation studies, they further bolstered their claims by showing that using motifs, intra-motifs connections and IP contribute towards the performance of the model.

**Weaknesses:**

1. While the architecture found by the model outperforms the other baselines, it comes at a significant a computational cost. Please report the time taken to run the search and the number of parameters of each model.

**Questions:**

1. Generally in DARTS, the best architecture found in the supernet is retrained from scratch. The accuracy of the retrained architecture is reported. Can the architecture found in your supernet be deployed as is?
2.  While the intra-motif connections are detailed, the search space of inter-motif connection is not elaborated in 3.2.1. Have I missed it?If not, can you please describe that too? Given that there are no explicit constrains in the search space formulation of the layer connection matrix, how can we enforce sparse inter-motif connections? In DARTS search space, a node i can only be connected with a node j if i < j. One can enforce a similar constraint in this case too.
3. Like you pointed out, the validation loss of the continuous representation is lower than the discretized version. So several works such as RobustDarts (that was cited in your papaer) suggested various regularization techniques to alleviate it. Similar to Robust Darts, can you also empirically show what the validation loss before and after descretization is? How does using SpiKL-IP influence it?
4. Can you also perform random search in your search space? In NAS, generally random search is also used as a baseline to understand the effectiveness of the proposed search algorithm.

---

> ### Author Response · Authors · 2023-11-20
> **Thanks for your insightful questions.**
>
> Q1: Yes, after we determine the optimal architecture, we will continue to use BP and IP to train on the optimal architecture for several epochs to adjust the weights. For example, for TI46-Alpha dataset, a complete training process of a RSNN with 800 neurons hidden layer for TI46-alpha dataset, including 150 epoch for cell size search, 150 epoch for connection type search and 400 epoch for finetune, takes 4 hours on single NVIDIA GeForce RTX 3090 GPU.
>
> Q2: The search space of inter-motif connections are intuitively illustrated in Figure 1\&2. In detail: 1. In terms of allowable connections and the number of such connections with in a SC-ML layer, inter-motif connections are manually restricted between adjacent motifs, and neurons in one motif are only allowed to be wired up with the corresponding neurons in the neighboring motifs (The i-th motif is only allowed to be connected to the i-1 and i+1 motifs). This means that with in a SC-ML layer, the inter/intra-motif connection of n neurons in space is local and sparse, reducing from the order of $O(n^2)$ to the order of $O(n)$, which also reduces its search space. 2. In terms of connection types, we allowed the type of of an inter-motif connection to be any one of the following three: inhibitory, excitatory or non-existent; and we determined the type of each inter-motif connection through search. Furthermore, inhibitory connections are set to fixed values and non-existent connections are set to 0, neither of which needs to be optimized.
>
> The aforementioned configuration was designed to allow flexibility in constructing a high-performance sparse spiking RNN, while simultaneously constraining the search space to ensure practical viability in solving the complex optimization problem associated with RNN architecture. The proposed optimization framework is highly flexible and can accommodate other types of sparsity constraints.
>
> Q3 & Q4:
> Please check general response/Section: Additional Information in Appendix of our paper. We provide comprehensive data & analysis of random search baseline, learning curve and ablation study of IP rule there.
>
> Thanks for your insightful question! We'd be glad to discuss any unclear details further.

---

### Author Response · Authors · 2023-11-20
**General Response**

We are grateful to the reviewers for their insightful and constructive suggestions. Based on this, we improved our work and revised the submitted version of the paper.  We provide additional data in section $\text{F. Additional information}$ in $\text{Appendix}$ of our paper, which is summarized as follows:

1.$\textbf{Additional data}$

We provided additional experimental data in Table below, including: random search in the search space as the baseline for HRMAS, effect of IP rule, HRMAS perfermence on Larger network. Experimental results show that: the HRMAS method shows consistent superiority (around 2\%) over the random search baseline; IP rule brings stable performance improvement (around 1.3\%); our method can be efficiently extended to networks with more neurons while providing good performance.


| Arch Optimization | Learning Rule | SC-ML Sizes | Best            | Mean           | Std        |
|-------------------|---------------|-------------|-----------------|----------------|------------|
| HRMAS (with IP)   | TSSL-BP       | 800         | **96.44%**      | **96.08%**     | **0.27%**  |
| HRMAS (with IP)   | TSSL-BP       | 1600        | **96.26%**      | -              | -          |
| HRMAS (with IP)   | TSSL-BP       | 2400        | **96.58%**      | -              | -          |
| HRMAS (with IP)   | TSSL-BP       | 3200        | **96.45%**      | -              | -          |
| HRMAS (w/o IP)    | TSSL-BP       | 800         | 95.17%          | 94.74%         | 0.32%      |
| Random            | TSSL-BP       | 800         | 94.47%          | 94.18%         | 0.30%      |

2.$\textbf{The computational resources required for our method}$

The proposed HRMAS bi-level optimization process is similar to DARTS, so the overall computational complexity is similar to DARTS; the IP method is an localized unsupervised learning method and does not constitute significant computational consumption. Furthermore, our proposed SC-ML topology greatly reduces the search space. Specifically, as the Figure9 in Appendix shows, the HRMAS optimization of a SC-ML layer, with $n$ neurons, a motif size of $s$ and $n/s$ motifs, reduces the parameters that need to be optimized for the recurrent connection matrix from $O(n^2)$ to $O(sn)$: $O(n/s)$ inter-motif connections + $O(n/s * s^2)$ intra-motif connections + $O(n)$ neuron hyperparameters. Generally, $s \ll n$, which reduces the parameter space of recurrent connections to linear growth with the neuron numbers, allowing our algorithm scale well. Specifically, a complete training process of a RSNN with 800 neurons hidden layer for TI46-alpha dataset, including 150 epoch for cell size search, 150 epoch for connection type search and 400 epoch for finetune, takes 4 hours on single NVIDIA GeForce RTX 3090 GPU.

3.$\textbf{IP rule's effect on performance during optimizing process}$

We plotted the performance curve of the network optimization process on the TI46-alpha dataset. Figure7 and Figure8 in the Appendix show the loss and accuracy on the validation set respectively. The solid line and shading show the mean and standard deviation of the 5 experiments. We conducted experiments with ip rule turned on and off. We use green text to mark each phase of architecture optimization.

The experimental results show: 1.When the network architecture changes drastically, such as iteration = 750 (the cell size search ends and the connection type search starts), and iteration = 1500 (the connection type search ends, the network is discretized and fine-tuned), the network There will be a slight performance degradation. But it can be quickly improved to a higher level by the next stage of training. 2.It can be found that IP method brings two benefits: improved network performance and a more stable training process. The figures showed that the red solid line (mean) has always performed better than the blue solid line without the IP method; at the same time, the red shadow (standard deviation) has always been narrower than the blue shadow, which means a more stable network architecture search process.

4.$\textbf{Sparsity recurrent connection of optimized RSNN}$

We shows the weight matrix of the RSNN with SC-ML optimized by the HRMAS method in the Figure9 in Appendix. The original fully connected recurrent matrix size is 800*800. We set the search space of motif size to [2,4,8,16,32]. In five random experiments, the HRMAS optimization method always gave the search results of motif-size=2, with similar inter/intra motif topology. This limits the huge recurrent matrix to a highly sparse band matrix with non-zero values only near the diagonal, greatly reducing the search space, parameter amount, and optimization difficulty.

---

### Meta-Review · Area_Chair_YjMs · 2023-12-05

**Metareview:**

The authors propose a recurrent spiking neural network architecture built upon sparsely connected recurrent motif layers (SC-MLs).  Each SC-ML layer consists of numerous recurrent motifs with sparse inter-connections.  The authors also introduce a so-called hybrid risk-mitigating architectural search (HRMAS) to automatically optimize the network topology which includes DARTS-like differentiable architectural search and a biologically inspired "self-repairing" based on intrinsic plasticity.  Experiments on four datasets demonstrate superior performance of the proposed HRMAS-optimized RSNN over peer RSNNs or SNNs.  Globally the paper is well written and the topic under investigation is important to the community.  Some of the concerns raised by the reviewers have been successfully cleared by the authors in their rebuttal (e.g. the computing complexity and training cost). That being said, there are some standing concerns after the rebuttal and discussion.  For example,  the proposed HRMAS is similar to DARTS.  The major difference is introducing SC-MLs and intrinsic plasticity which are very specialized to SNNs.  So the novelty is not overwhelmingly significant.  Also,  it is not clear how SOTA NAS approaches would perform in this case. It would be helpful to investigate it to understand where HRMAS really stands.  Despite an interesting study, the paper needs to make a stronger case.   I would suggest the authors improve the paper according to the reviews and submit it to a future venue.

**Justification For Why Not Higher Score:**

The novelty is not significant considering the similarity with DARTS.

**Justification For Why Not Lower Score:**

N/A

---

### Decision · Program_Chairs · 2024-01-16

Reject